# Machine learning for early prediction of the infection in patients with urinary stone after treatment of holmium laser lithotripsy

**Weiqi Xia[1], Miaomiao Zhang[2], Xiaowei Zheng[1], Zushuai Wu[1], Zixue Xuan[1], Ping Huang[1]\*, Xiuli Yang[1]\***

**1** Department of Pharmacy, Center for Clinical Pharmacy, Cancer Center, Zhejiang Provincial People's Hospital(Affiliated People's Hospital), Hangzhou Medical College, Hangzhou, Zhejiang, China, **2** School of Pharmacy, Hangzhou Normal University, Hangzhou, Zhejiang, China

\* xiuli8245@163.com (XY); huangping@hmc.edu.cn (PH)

## Abstract

Patients after holmium laser lithotripsy have a certain probability of getting postoperative infection. An early and accurate diagnosis of postoperative infection allows a timely administration of appropriate antibiotic treatment. However, doctors can not accurately determine whether the patient has the infection. Here, a novel strategy is put forward to assist in predicting postoperative infection early by using machine learning methods. We retrospectively collected 1006 cases of patients with urinary stone after treatment of holmium laser lithotripsy from Zhejiang Provincial People's Hospital. Feature engineering was added to filter the important characteristics and Miceforest multiple imputation method was applied to tackle the missing data problem. We used 5-fold cross-validation to train and validate the six machine learning methods. Besides, we could also find key variables important to postoperative infection by explaining the model. The hyperparameters were constantly adjusted to achieve the best performance of model. The result showed that LR had a best performance in independent datasets with AUC of 0.734. And the SHAP values indicated that preoperative urine leukocyte count was the most important variable to the prediction. Our study enables accurate predictions of infection in urology perioperative periods, the key variables can be interpreted better and more accurately to support clinical decision making.

## Introduction

Urinary tract stone is one of the most common diseases of urinary system, and the prevalent rate of the population is about 1%–15%, among which upper urinary tract stones is about 70 percent [1]. Holmium laser lithotripsy is used for the clinical treatment of upper urinary tract stones [2]. However, it still results in a high incidence of postoperative infection and has the risk of the progression to urosepsis in the later stage (about 12–20%) [3,4]. Stone location, stone size and so on could affect the

**Data availability statement:** Data was approved for limited use by the Institutional Review Boards of the respective hospitals, and not publicly available.

**Funding:** This work was financially supported by Research Program for Medicine and Health Science and Technology of Zhejiang Province (2022KY585 to X. Yang).

**Competing interests:** The authors have declared that no competing interests exist.

surgery [5]. If urosepsis is not controlled timely and effectively, the condition may rapidly progress to septic shock [6], and eventually leading to multiple organ failure, with a fatality rate of up to 22%~76%, which will not only bring great pain to the patients themselves, but also bring economic burden to the family and society. Therefore, early identification of high-risk patients complicated with infection after holmium laser lithotripsy and timely adoption of targeted intervention measures can not only reduce the probability of pyelonephritis and risk factors for septic shock, but also promote the rehabilitation of patients. Currently, the occurrence of postoperative infections is typically identified through the clinical experience of the doctors or by monitoring infection indicators. We hope to leverage machine learning methods to detect infection earlier, thereby avoiding the overuse of antibiotics.

Machine learning (ML), as a branch of artificial intelligence (AI), uses computer systems to realize prediction or decision-making tasks with algorithms and statistical models, which can provide great possibilities for the progress of medical science [7]. Machine learning is playing an increasingly important role in the medical field because of its methodological advantages, such as cardiovascular disease prediction [8–10], cancer prediction [11,12], adverse reaction prediction [13,14] and so on. Therefore, in the clinical field, the use of machine learning methods to mine medical big data, identify valuable features, and build prediction models adapted to complex data is a current research hotspot. Unlike traditional methods that depend on postoperative infection indicators to identify infections, machine learning methods can predict the likelihood of postoperative infections beforehand, which enables the appropriate and timely administration of antibiotics, thereby reducing the overall use of antibiotics and minimizing the risk of antibiotic resistance.

In this study, six machine learning prediction models were established to determine whether postoperative infection occurred in urology perioperative periods [15]. Firstly, 1006 cases of holmium laser lithotripsy in Zhejiang Provincial People's Hospital from October 2018 to August 2021 were collected in this study. Preoperative and intraoperative risk factors of patients were collected, and Chi-square test [16] was used for feature selection. Since some variable factors contained missing values, we applied multiple imputation to make up for the missing data [17,18]. Secondly, the prediction models were constructed by using the six mainstream machine learning methods, and the hyperparameters were constantly adjusted to select the best model. Besides, we used the 5-fold cross-validation to ensure the generalization performance of the model. We also showed the importance of those characteristics of patients in medical records. The SHapley Additive exPlanation (SHAP) values [19] were used to explain these prediction models. Our study enables accurate predictions of infection in urology perioperative periods, the interpretation of key variables can be interpreted better and more accurately to support clinical decision making.

## Materials and methods

### Data collection

A total of 1006 patients with upper urinary calculi who received treatment in Zhejiang Provincial People's Hospital from October 2018 to August 2021 were collected in this

study. The dates from a retrospective study were accessed for research purposes on Mar27, 2023 by Ethics Committee of the Zhejiang Provincial People's Hospital. The incidence of postoperative infection in 1006 patients after surgery was observed. Postoperative infections include postoperative urinary tract infection, systemic inflammatory response syndrome, and urosepsis, the diagnostic criteria were based on EAU Guidelines on Urological Infections (2020). Finally, 325 patients were infected in urology perioperative periods and 681 were not infected. The characteristics of the patients, such as age, gender, hydronephrosis, hypertension, diabetes, nephrological related diseases, preoperative infection, stone location, intraoperative abnormalities (such as turbid urine and ureter malformation during the operation), operation duration were recorded and analyzed for the following process of modeling and analysis.

## Data preprocessing

In order to construct the prediction model, the above variables were converted into numerical values in this study. For the convenience of operation, each index was assigned as a binary variable. The details are in the S1 File.

## Feature selection

In the process of machine learning model construction, feature engineering refers to the screening and conversion of features extracted from original data into suitable subset, and the subset of these characteristics is the input of the model [20]. In this study, the feature selection method based on Chi-square test [21] was used to filter the above features. Chi-square test is an independence test for discrete variables. The null hypothesis of Chi-square test is that two discrete variables are independent of each other. In feature selection, it is used to distinguish labels and features, so it can be used to judge whether a certain feature and label are independent. If so, it means that the feature is not helpful to the prediction of labels. So in many cases, the chi-square test is a very important way to eliminate irrelevant features. The formula for the chi-square test is as follows:

$$\chi^2 = \sum_{i=1}^{k} \frac{(A_i - np_i)^2}{np_i} \tag{1}$$

## Imputation of missing data

Multiple Imputation is a method to deal with missing values based on repeated simulation [22], which can provide a more accurate estimate of missing data compared with single mean or mode imputation. Here, in this paper, we apply the multiple imputation method Miceforest to impute the missing data.

## Model construction

**SVM.** Support vector machine (SVM) [23] is a kind of classification model. Its principle is to find a hyperplane in the space which can divide the data into different parts. For the binary classification problem, SVM algorithm needs to find parameters $w$ and $b$ to divide the positive and negative samples well.

**LR.** Logistic regression (LR) [24] is a linear classifier. Through logistic function (Sigmoid function), the data features are mapped to a probability value (the possibility that the sample belongs to a positive example) in the interval of 0~1.

**RF.** Random Forest (RF) is an ensemble algorithm [25], which integrates multiple decision tree sets, but there is no direct correlation among decision trees. Each decision tree in the random forest classifies randomly selected samples according to the characteristic attributes of the selected samples. Each decision tree has a classification result, and finally according to the prediction results of these trees, the RF votes to decide the final prediction category of the samples.

**DT.** Decision Tree (DT) is a tree structure [26], in which each internal node represents a judgment on an attribute, each branch represents the output of a judgment result, and finally each leaf node represents a classification result. Essentially, DT classifies datasets through several conditional discriminant procedures and finally obtains the desired results.

## XGBoost、LightGBM

eXtreme Gradient Boosting (XGBoost) and Light Gradient Boosting Machine (LightGBM) are all boosting algorithms [27,28]. Gradient Boosting Decision Tree (GBDT) is the basis of XGBoost and LightGBM. In each iteration, GBDT calculates the current negative gradient of the sample based on the empirical loss function, and then a new weak classifier is trained and the weight of the weak classifier is calculated to update the model. The difference between GBDT and XGBoost is that XGBoost adds regularization method. The formula is as 2. LightGBM is an improved version of XGBoost. Compared to XGBoost, it adds many new methods to improve the model, including parallel schemes, gradient-based unilateral detection, and so on. Technical details are given in the S1 File.

$$Obj = \sum_{i=1}^{m} I(y_i, \hat{y}_i) + \sum_{k=1}^{K} \Omega(f_k) \tag{2}$$

## Evaluation

A common criterion for evaluating binary or multi-classification models is confusion matrix, which compares the predicted label with real label [29]. True Positive (TP) means that both the true label of the sample and the predicted label of model are positive. False Negative (FN) represents that the positive sample is predicted incorrectly, while False Positive (FP) refers to the negative sample is considered as positive. True Negative (TN) means that both the true label of the sample and the prediction of the model are negative. The accuracy, precision, recall can be obtained by calculating the prediction label and the real label of the model according to these four indexes. According to the precision and recall, we can calculate the F1. Area under the receiver operator characteristics curve (AUROC) is also used as the evaluation metric.

$$Accuracy = (TP + TN)/(TP + TN + FP + FN) \tag{3}$$

$$Precision = TP/(TP + FP) \tag{4}$$

$$Recall = TP/(TP + FN) \tag{5}$$

$$F1 = 2 * (precision * recall) / (precision + recall) \tag{6}$$

## Results

### Statistical analysis

The datasets had 14 characteristics and the targets were postoperative infection or no infection. Table 1 showed the all characteristics of the patient, and we used Chi-square test to analyze the feature set. When P-value is less than 0.05, it indicates that there is a significant difference between the two groups of data in this feature. Therefore, the features with P-value less than 0.05 were included in the following model training. It could be seen from Table 1 that the P-value of a total of 8 indicators were less than 0.05, including Age, Preoperative albumin, Duration of surgery, Asymptomatic bacteriuria, The control of preoperative infection, Intraoperative abnormalities, Preoperative urine leukocyte count and Urine occult blood.

### Hyperparameters optimization

We used 5-fold cross-validation to train and test the model. The dataset was divided into five parts. One of the five sets was selected as test set, the rest four sets were selected as training set. To optimize each machine learning models,

**Table 1. Characteristics of all urinary stone patients and the result of Chi-square test.**

| Characteristic | | Assignment | postoperative infection in urology (32) | No infection (681) | χ² | Pvalue |
|---|---|---|---|---|---|---|
| Sex | Male | 1 | 96 | 185 | 0.5030 | 0.4782 |
| | Female | 0 | 229 | 496 | | |
| Age | >=60 | 1 | 236 | 447 | 4.5976 | 0.0320 |
| | <60 | 0 | 89 | 234 | | |
| Hydronephrosis | Yes | 1 | 251 | 505 | 0.9554 | 0.3283 |
| | No | 0 | 74 | 176 | | |
| High blood pressure | Yes | 1 | 94 | 188 | 0.1294 | 0.7190 |
| | No | 0 | 231 | 493 | | |
| Preoperative albumin | <40g/L | 1 | 127 | 149 | 31.8246 | <0.0001 |
| | >=40g/L | 0 | 198 | 532 | | |
| Diabetes | Yes | 1 | 45 | 83 | 0.4057 | 0.5242 |
| | No | 0 | 280 | 598 | | |
| Nephrological related diseases | Yes | 1 | 83 | 152 | 1.0994 | 0.2944 |
| | No | 0 | 242 | 529 | | |
| Duration of surgery | >=60min | 1 | 119 | 163 | 16.9110 | <0.0001 |
| | <60min | 0 | 206 | 518 | | |
| Location of kidney stone | Kidney | 1 | 147 | 277 | 1.6901 | 0.1936 |
| | Ureter | 0 | 178 | 404 | | |
| Asymptomatic bacteriuria | Yes | 1 | 280 | 458 | 39.2504 | <0.0001 |
| | No | 0 | 45 | 223 | | |
| The control of Preoperative infection | Yes | 1 | 102 | 165 | 5.4165 | 0.0199 |
| | No | 0 | 223 | 516 | | |
| Intraoperative abnormalities | Yes | 1 | 261 | 465 | 15.2471 | 0.0001 |
| | No | 0 | 64 | 216 | | |
| Preoperative urine leukocyte count | >= 25 | 1 | 67 | 43 | 44.7461 | <0.0001 |
| | <25 | 0 | 258 | 638 | | |
| Urine occult blood | Yes | 1 | 277 | 548 | 3.0647 | 0.0080 |
| | No | 0 | 48 | 133 | | |

different hyperparameters were explored through grid search. In LR, we selected regularization method with L1, L2, penalty coefficient C with 0.001, 0.01, 0.1, 1, 10, 100, 1000. In SVM, different kernel function with linear, poly, rbf, penalty coefficient C with 1, 10, 100, 1000 were included. For DT, we considered the maximum depth with 30, 50, 60, 100, min_samples_leaf with 2, 3, 5, 10 and min_impurity_decrease with 0.1, 0.2, 0.5. The n_estimators of RF ranged from 10 to 100. For XGBoost, we considered the maximum depth from 1 to 20, min_child_weight from 1 to 20 and n_estimators from 100 to 200. And for LightGBM, we used the maximum depth with 5, 10, 15, 20, 30 and the learning rate with 0.1, 0.2, 0.3, 0.4, 0.5, 0.6, 0.7, 0.8. The best hyperparameters were selected according to the mean performance based on cross validation [30,31]. And the optimum values were displayed in Table 2.

**Imputation method analysis**

We compared Multiple Imputation method with Mean Imputation method by train six machine learning models. Table 3 was the AUC result of two different imputation methods. We could clearly find that the prediction ability of Multiple Imputation method was greater than Mean Imputation.

**Table 2. Optimization of model based on grid search.**

| Model | Hyperparameters | Values | Optimum value |
|---|---|---|---|
| LR | regularization method | L1,L2 | L2 |
| | C | 0.001, 0.01, 0.1, 1, 10, 100, 1000 | 0.01 |
| SVM | kernel | linear, poly, rbf | rbf |
| | C | 1, 10, 100, 1000 | 1 |
| DT | max_depth | 30, 50, 60, 100 | 30 |
| | min_samples_leaf | 2, 3, 5, 10 | 2 |
| | min_impurity_decrease | 0.1, 0.2, 0.5 | 0.1 |
| RF | n_estimators | 10–100 | 90 |
| XGBoost | max_depth | 1–20 | 1 |
| | min_child_weight | 1–20 | 1 |
| | n_estimators | 10–200 | 10 |
| LightGBM | max_depth | 5, 10, 15, 20, 30 | 10 |
| | learning_rate | 0.1, 0.2, 0.3, 0.4, 0.5, 0.6, 0.7, 0.8 | 0.1 |

**Table 3. The comparison between mean imputation and multiple imputation.**

| Method | Mean Imputation | Multiple Imputation |
|---|---|---|
| LR | 0.679 | 0.701 |
| SVM | 0.628 | 0.657 |
| DT | 0.500 | 0.500 |
| RF | 0.684 | 0.714 |
| XGBoost | 0.680 | 0.710 |
| LightGBM | 0.640 | 0.687 |

## Performance evaluation after normalization

The purpose of normalization is to limit the feature in a certain range so that we can eliminate the adverse effects caused by the singular sample data. We normalized the continuous variables described above. SVM, RF, LR, DT, XGBoost and LightGBM were respectively used in this study. During the training, 5-fold cross-validation was adopted, and the minimum loss was taken as the training objective. Fig 1 showed the AUC results of SVM, LR, DT, RF, LightGBM and XGBoost models, where each line represented a validation in the 5-fold cross-validation. As can be seen from the figure, except DT, the average AUC of the model was greater than 0.70.

The above 5-fold cross-validation results indicated that the DT model had no classification ability at all in this dataset, so we only compared the performance of the other five models. Fig 2 displayed the performance of the independent test dataset based on the five classification models. In the figure, the horizontal axis represented AUC, Accuracy, F1 respectively, while the vertical axis represented the corresponding value. The green bar was SVM, the orange was LR, the grey bar was RF, the red was LightGBM, and the blue was XGBoost. It could be clearly seen that the performance of LR and RF on AUC was superior than other models. But in accuracy and F1, LightGBM performed the best. Positive Predictive Value (PPV) indicates the proportion of patients who are judged as positive (infected) by the test that are actually infected. It serves as a metric for assessing the test's capability to avoid "false positive" misdiagnoses. The Positive Predictive Values (PPVs) of these five models are 0.687, 0.73, 0.712, 0.733, 0.697 respectively. LightGBM performed the best.

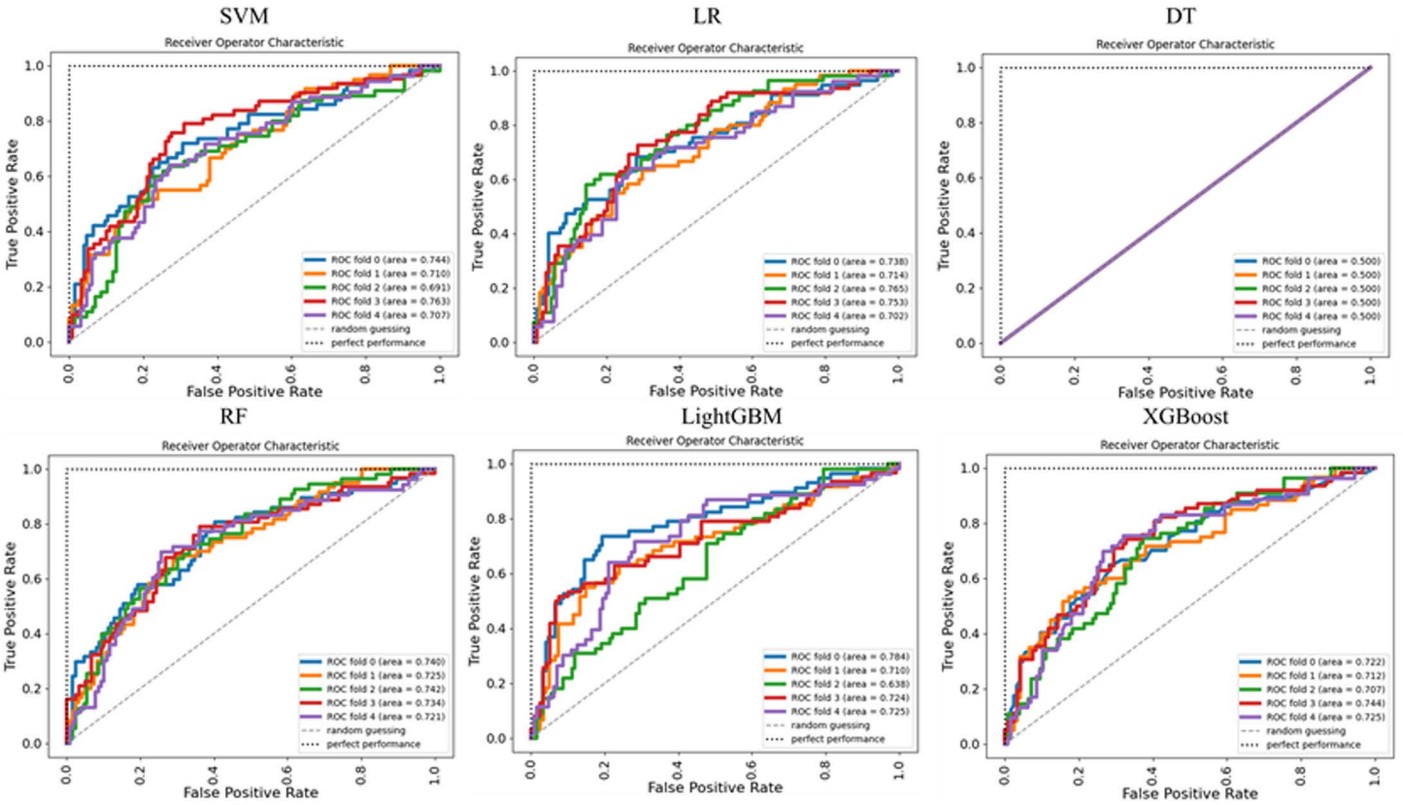

**Fig 1. 5-cross validation results of six models.**

## Importance of clinical variables

In order to facilitate the interpretation of model, the SHAP value has been introduced to judge the positive or negative relationship of clinical variable with prediction [32]. A dot is created for each feature attribution value for the model of each patient, and thus one patient is allocated one dot on the line for each feature. Red represents higher feature values, and blue represents lower feature values. The ranking of the features is in descending order according to the average absolute value of SHAP. RF model was chosen to analyze the importance of characteristics. According to the importance and impacts of variables on model prediction, a bee swarm plot was formed for each feature. As shown in Fig 3, In RF, we found that patients with high preoperative urine leukocyte count (red) had a higher risk of developing postoperative infection than patients with low preoperative urine leukocyte count (blue). Other characteristics were also found that a high value had positive effect on developing postoperative infection.

## Discussion

Perioperative postoperative infection is an infection that occurs within 1 month after surgical operation or surgical intervention [33,34], which is one of the common complications after surgical operation. Individuals suffering from ureteral stones often experience persistent, low-grade inflammation within the ureteral lining as a result of the prolonged mechanical irritation caused by the stone. These patients are prone to getting perioperative postoperative infection which can lead to prolonged time in hospital, increased medical costs, increased mortality and a variety of adverse outcomes. Therefore, for subsequent antibiotic administration and patient's life safety, it is vital to predict whether patients have a risk of infection [35]. If we can detect patients with infection as early as possible, then the clinician can make a more rational medication.

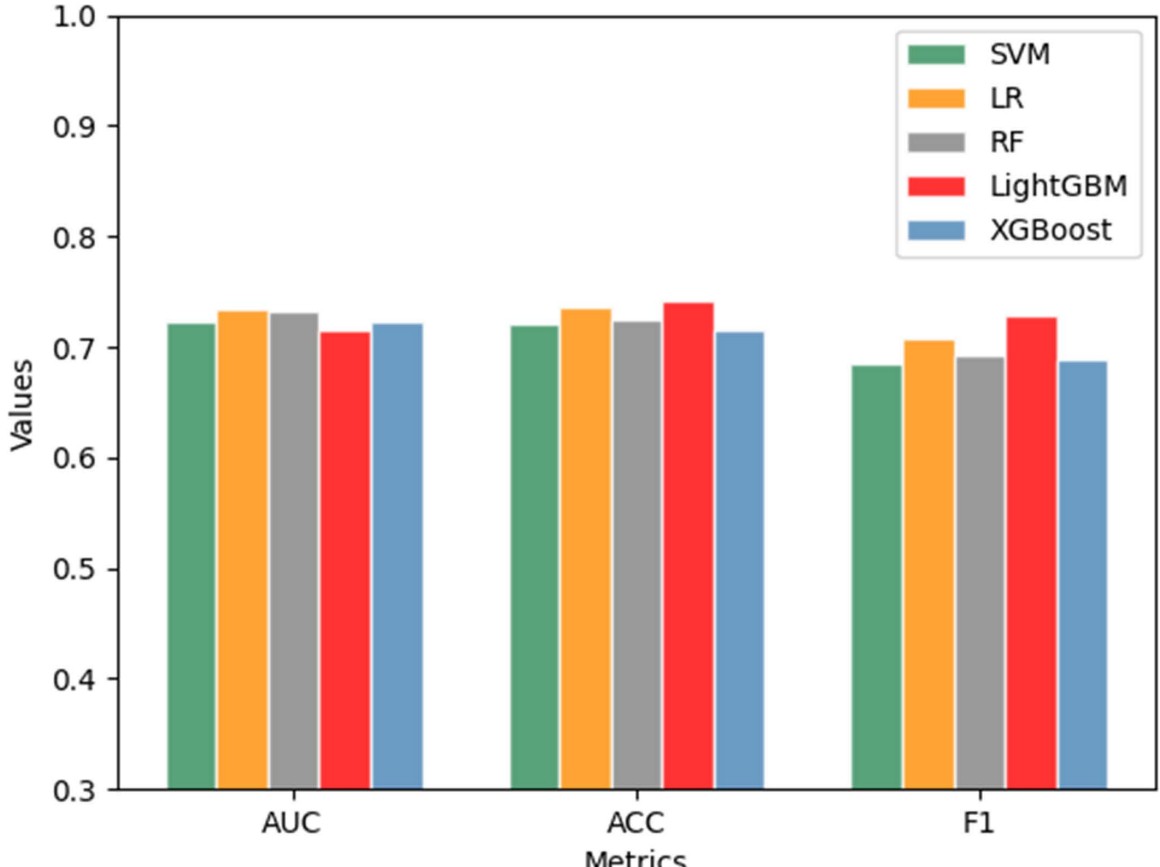

**Fig 2. Performance evaluation of five machine learning methods.**

Several models have been developed for the prediction of infection after urolithiasis surgery [36,37]. Nonetheless, the postoperative infection risk factors considered in previous studies are not comprehensive. Here, we developed an early prediction of the infection in patients with urinary stone after treatment of holmium laser lithotripsy by applying machine learning methods to help physicians accurately identify the tendency of patients to get infected after surgery. By selecting the best dataset imputation method and the best number of clinical variables (eight of fourteen variables were chosen in our risk prediction model), and constantly adjusting the model parameters, we constructed the model with a reliable ability to predict postoperative infection. We used six mainstream machine learning methods and the grid seach method to find the best parameters, and the LR model performed best in the AUC value.

The results of this study demonstrate that preoperative urine leukocyte count and asymptomatic bacteriuria are significantly associated with the occurrence of postoperative infection. Numerous other studies have also shown that preoperative elevated urine leukocyte counts, positive urine cultures, and positive nitrite tests in urine are significant risk factors for postoperative infections, with the degree of bacteriuria potentially correlating to the severity of postoperative urinary tract infection [38,39]. Besides, Preoperative albumin, Duration of surgery, the presence of urinary tract anomalies discovered during surgery are associated with an increased risk of postoperative infections [36]. Of course, there are some limitations in this study. This is a single-center study, which limits the generalizability of our findings. Compared to single-center studies, multi-center studies involve a more geographically and demographically diverse patient population, providing a broader representation of the target population. Thus, in the future work, our model should be further optimized in

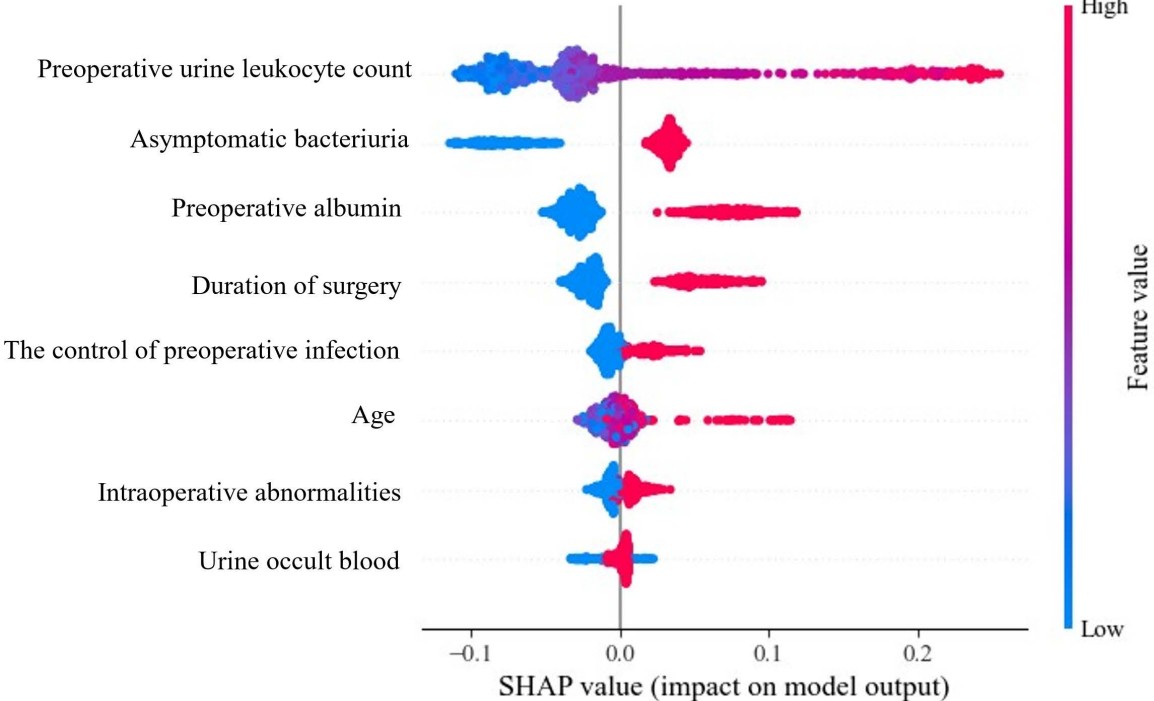

**Fig 3. RF model feature importance explained by SHAP values.**

multi-centric real world clinical data and other public clinical database. In the process of machine learning model construction, the feature selection method is not very comprehensive. In the future research, more feature selection methods should be tried. Besides, we should consider more variables vital to the prediction of postoperative infection in order to further improve the performance of model.

## Conclusion

At present, clinicians frequently administer antibiotics without assessing the actual infection status of the patients, resulting in the overuse of antibiotics. However, determining the likelihood of postoperative infections places an additional burden on clinicians and typically requires the expertise of experienced clinicians. To address these challenges, this study employs machine learning methods to develop a postoperative infection prediction tool which can assist clinicians in predicting infections in urology perioperative periods, thereby significantly reducing the unnecessary use of antibiotics and improving overall patient care. The findings may help doctors guide patient treatment, and prevent unnecessary pain and cost. To enhance the usability of this tool, we plan to develop it into a mobile application. This app will enable doctors to input various characteristics directly, and the app will instantly provide a prediction of the likelihood of postoperative infection. This user-friendly approach will facilitate more accurate and timely decision-making, ultimately improving patient care and reducing the overuse of antibiotics.

## Supporting information

**S1 File. The S1 File includes the detail of Data preprocessing and Technical Details for the Statistical Analysis.** (DOCX)

## Acknowledgments

This work was financially supported by Research Program for Medicine and Health Science and Technology of Zhejiang Province (2022KY585 to X. Yang)

## Author contributions

**Conceptualization:** Xiaowei Zheng, Xiuli Yang.

**Data curation:** Miaomiao Zhang, Zushuai Wu, Zixue Xuan.

**Formal analysis:** Weiqi Xia.

**Writing – original draft:** Weiqi Xia.

**Writing – review & editing:** Weiqi Xia, Ping Huang, Xiuli Yang.

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
