## [Decision Letter · Decision Letter 0]

6 Nov 2024

PONE-D-24-39210Machine learning for early prediction of the infection in patients with urinary stone after treatment of holmium laser lithotripsyPLOS ONE

Dear Dr. Yang,

Thank you for submitting your manuscript to PLOS ONE. After careful consideration, we feel that it has merit but does not fully meet PLOS ONE’s publication criteria as it currently stands. Therefore, we invite you to submit a revised version of the manuscript that addresses the points raised during the review process.

We look forward to receiving your revised manuscript.

Kind regards,

Bryan Kwun-Chung Cheng

Academic Editor

PLOS ONE

Journal Requirements:

5. We note that you have indicated that there are restrictions to data sharing for this study. PLOS only allows data to be available upon request if there are legal or ethical restrictions on sharing data publicly. For more information on unacceptable data access restrictions, please see http://journals.plos.org/plosone/s/data-availability#loc-unacceptable-data-access-restrictions.

Reviewers' comments:

Reviewer's Responses to Questions

**Comments to the Author**

1. Is the manuscript technically sound, and do the data support the conclusions?

Reviewer #1: Yes

Reviewer #2: Yes

2. Has the statistical analysis been performed appropriately and rigorously? 

Reviewer #1: Yes

Reviewer #2: Yes

3. Have the authors made all data underlying the findings in their manuscript fully available?

Reviewer #1: No

Reviewer #2: Yes

4. Is the manuscript presented in an intelligible fashion and written in standard English?

Reviewer #1: Yes

Reviewer #2: Yes

5. Review Comments to the Author

Reviewer #1: This paper used multiple machine learning algorithms trying to predict the risk of infection after treatment of holmium laser lithotripsy in a clinical setting. The topic is of interest and the analysis has a relatively complete workflow process. However, there still are some concerns about the details:

1. In the first paragraph of Introduction, the authors should not only introduce the prevalence of urinary tract stone, but also the epidemiology of the infection after holmium laser lithotripsy, such as its rate and risk factors.

2. In Data preprocessing section of Materials and methods, the variable coding methods can be moved into supplementary file, and it’s better to include only feature engineering methods in this part like data transformation and data extraction, if any.

3. Please give the names and versions of all the packages used in each step of the analyses.

4. Do you have both a validating set and a test set? If not, how to identify the best hyperparameters and also evaluate the model performance using the only one testing set under a 5-CV strategy?

5. Usually, in clinical predictive models, PPV is very low even if other performance indexes are relatively high, so do you consider additionally calculate PPV to evaluate your model’s performance?

6. In the Discussion, lacking external validation may be one of the limitations.

Reviewer #2: I have carefully reviewed the manuscript titled "Machine Learning for Early Prediction of the Infection in Patients with Urinary Stone After Treatment of Holmium Laser Lithotripsy." The study focuses on utilizing machine learning models to predict postoperative infections, which is an important step in improving clinical outcomes and personalizing treatment for patients undergoing urinary stone surgery. I commend the effort put into this work and would like to provide the following feedback:

Content and Structure: The manuscript is well-structured, providing a clear explanation of the problem, methodology, and results. However, the introduction would benefit from further elaboration on the clinical challenges of postoperative infection prediction and how machine learning models can provide a unique solution compared to traditional methods. Additionally, the conclusion should expand on the broader implications of the findings and how they can be applied to real-world clinical settings.

Literature Review and Citations: The literature review is comprehensive, but incorporating more recent studies on machine learning applications in medical prediction models would strengthen the manuscript. I recommend adding the following papers:

https://doi.org/10.1016/j.eswa.2023.122147

https://doi.org/10.1007/s10586-023-04221-5

https://doi.org/10.54216/JAIM.080103

These references will help contextualize your research within the broader landscape of recent advancements in medical prediction and machine learning.

Technical Clarifications and Suggestions: While the technical aspects of the model are well-described, further clarification on the hyperparameter tuning process for the machine learning models would provide valuable insights. Specifically, more details on how the models were optimized (e.g., grid search, cross-validation) would enhance the reproducibility of the study. Additionally, a more detailed comparison of the different models (e.g., Random Forest, Logistic Regression, XGBoost) in terms of computational efficiency and predictive power could provide a deeper understanding of their strengths and weaknesses in this context.

Lastly, it would be helpful to discuss potential limitations of the study, such as the single-center nature of the dataset, and how these limitations might affect the generalizability of the model.

I hope these suggestions assist in refining your manuscript. The research is timely and relevant, and with these revisions, it could make a valuable contribution to the field of predictive analytics in healthcare.

6. PLOS authors have the option to publish the peer review history of their article (what does this mean? ). If published, this will include your full peer review and any attached files.

**Do you want your identity to be public for this peer review?** For information about this choice, including consent withdrawal, please see our Privacy Policy .

Reviewer #1: **Yes: ** JUN HE

Reviewer #2: **Yes: ** Abdelaziz Abdelhamid

---

## [Author Response · Author response to Decision Letter 1]

3 Dec 2024

Dear Reviewers,

Thank you so much for your comments concerning our manuscript entitled “Machine learning for early prediction of the infection in patients with urinary stone after treatment of holmium laser lithotripsy” (ID: PONE-D-24-39210). Those comments are all valuable and very helpful for revising and improving our paper, as well as the important guiding significance to our researches. We have studied comments carefully and have made correction which we hope meet with approval.

Reviewer #1:

1. In the first paragraph of Introduction, the authors should not only introduce the prevalence of urinary tract stone, but also the epidemiology of the infection after holmium laser lithotripsy, such as its rate and risk factors.

Authors' Response: Thanks for the kind and insightful comments and suggestions! We have re-written this part according to your suggestion. Changes to our manuscript were all highlighted within the document by using red-colored text.

2. In Data preprocessing section of Materials and methods, the variable coding methods can be moved into supplementary file, and it’s better to include only feature engineering methods in this part like data transformation and data extraction, if any.

Authors' Response: Thank you very much for your advice. We have removed the variable coding methods into supplementary file.

3. Please give the names and versions of all the packages used in each step of the analyses.

Authors' Response: Thank you very much for your advice. We have added the details of packages into supplementary file.

4. Do you have both a validating set and a test set? If not, how to identify the best hyperparameters and also evaluate the model performance using the only one testing set under a 5-CV strategy?

Authors' Response: Thank you very much for your advice. We have divided the datasets into 2 parts, one was used for 5-cv training process for the best hyperparameters, the other one was regarded as independent test dataset. Figure 1 was 5-cross validation results. We saved the best model to predict the independent dataset as showed in Figure 2.

5. Usually, in clinical predictive models, PPV is very low even if other performance indexes are relatively high, so do you consider additionally calculate PPV to evaluate your model’s performance?

Authors' Response: Thank you very much for your advice. Positive Predictive Value (PPV) is equivalent to precision. We had included the F1 score, which simultaneously evaluated both precision and recall. A high F1 score indicates that the model has both good precision and good recall.

6. In the Discussion, lacking external validation may be one of the limitations.

Authors' Response: Thank you very much for your advice. In fact, we have recently collected more data and are also attempting to use our developed model to predict whether patients will get infections after holmium laser lithotripsy surgery in order to give antibiotic intervention. Our model should be further optimized according to more datasets.

Reviewer #2:

1. Content and Structure: The manuscript is well-structured, providing a clear explanation of the problem, methodology, and results. However, the introduction would benefit from further elaboration on the clinical challenges of postoperative infection prediction and how machine learning models can provide a unique solution compared to traditional methods. Additionally, the conclusion should expand on the broader implications of the findings and how they can be applied to real-world clinical settings.

Authors' Response: We feel great thanks for your professional review work on our article. We have refined our introduction and the conclusion according to your suggestions. Changes to our manuscript were all highlighted within the document by using red-colored text.

2. Literature Review and Citations: The literature review is comprehensive, but incorporating more recent studies on machine learning applications in medical prediction models would strengthen the manuscript. I recommend adding the following papers:

Authors' Response: Thank you very much for your advice. We have added the papers in our study.

3. Technical Clarifications and Suggestions: While the technical aspects of the model are well-described, further clarification on the hyperparameter tuning process for the machine learning models would provide valuable insights. Specifically, more details on how the models were optimized (e.g., grid search, cross-validation) would enhance the reproducibility of the study. Additionally, a more detailed comparison of the different models (e.g., Random Forest, Logistic Regression, XGBoost) in terms of computational efficiency and predictive power could provide a deeper understanding of their strengths and weaknesses in this context.

Authors' Response: Thank you very much for your advice. We have added the details of training process into supplementary file. Due to the nature of algorithm, tree algorithms like XGBoost and LightGBM take longer time and consume more power, and Logistic Regression takes the minimum time.

4. Lastly, it would be helpful to discuss potential limitations of the study, such as the single-center nature of the dataset, and how these limitations might affect the generalizability of the model.

Authors' Response: Thank you very much for your advice. We have added the discussion in our paper in the Discussion part.

Yours sincerely,

Xiuli YANG

---

## [Decision Letter · Decision Letter 1]

25 Dec 2024

PONE-D-24-39210R1Machine learning for early prediction of the infection in patients with urinary stone after treatment of holmium laser lithotripsyPLOS ONE

Dear Dr. Yang,

Thank you for submitting your manuscript to PLOS ONE. After careful consideration, we feel that it has merit but does not fully meet PLOS ONE’s publication criteria as it currently stands. Therefore, we invite you to submit a revised version of the manuscript that addresses the points raised during the review process.

We look forward to receiving your revised manuscript.

Kind regards,

Bryan Kwun-Chung Cheng

Academic Editor

PLOS ONE

Journal Requirements:

Reviewers' comments:

Reviewer's Responses to Questions

**Comments to the Author**

1. If the authors have adequately addressed your comments raised in a previous round of review and you feel that this manuscript is now acceptable for publication, you may indicate that here to bypass the “Comments to the Author” section, enter your conflict of interest statement in the “Confidential to Editor” section, and submit your "Accept" recommendation.

Reviewer #1: (No Response)

Reviewer #2: All comments have been addressed

2. Is the manuscript technically sound, and do the data support the conclusions?

Reviewer #1: Yes

Reviewer #2: Yes

3. Has the statistical analysis been performed appropriately and rigorously? 

Reviewer #1: Yes

Reviewer #2: Yes

4. Have the authors made all data underlying the findings in their manuscript fully available?

Reviewer #1: No

Reviewer #2: Yes

5. Is the manuscript presented in an intelligible fashion and written in standard English?

Reviewer #1: Yes

Reviewer #2: Yes

6. Review Comments to the Author

Reviewer #1: It's still suggested to calculate PPV although you have reported F1. This performance index could give audience a more comprehensive understanding of the model prediction, and it will not be a big issue if the index is relatively low because it's hard to get both a ideal recall and PPV in the same clinic predition model.

Reviewer #2: (No Response)

7. PLOS authors have the option to publish the peer review history of their article (what does this mean? ). If published, this will include your full peer review and any attached files.

**Do you want your identity to be public for this peer review?** For information about this choice, including consent withdrawal, please see our Privacy Policy .

Reviewer #1: No

Reviewer #2: **Yes: ** I have no competing interests.

---

## [Author Response · Author response to Decision Letter 2]

26 Dec 2024

1.It's still suggested to calculate PPV although you have reported F1. This performance index could give audience a more comprehensive understanding of the model prediction, and it will not be a big issue if the index is relatively low because it's hard to get both a ideal recall and PPV in the same clinic predition model.

Authors' Response: Thanks for the kind and insightful comments and suggestions! We have added the PPV index to evaluate our model. Changes to our manuscript were all highlighted within the document by using red-colored text.

---

## [Editor Report · Decision Letter 2]

2 Jan 2025

Machine learning for early prediction of the infection in patients with urinary stone after treatment of holmium laser lithotripsy

PONE-D-24-39210R2

Dear Dr. Xiuli Yang,

We’re pleased to inform you that your manuscript has been judged scientifically suitable for publication and will be formally accepted for publication once it meets all outstanding technical requirements.

Kind regards,

Bryan Kwun-Chung Cheng

Academic Editor

PLOS ONE
---

## [Editor Report · Acceptance letter]

PONE-D-24-39210R2

PLOS ONE

Dear Dr. Yang,

I'm pleased to inform you that your manuscript has been deemed suitable for publication in PLOS ONE. Congratulations! Your manuscript is now being handed over to our production team.

Kind regards,

on behalf of

Dr. Bryan Kwun-Chung Cheng

Academic Editor

PLOS ONE